# OpenReview forum: "Shared Global and Local Geometry of Language Model Embeddings"
_colmweb.org/COLM/2025/Conference — COLM 2025_

### Official Review · Reviewer_PyR3 · 2025-05-12

**Rating:** 8
**Confidence:** 3
**Ethics Flag:** 1

**Summary:**

This work primarily focuses on understanding the global geometry of tokens when considering the embeddings of pretrained LLMs. First the authors provide evidence as to how the embeddings of tokens have similar orientation (for embedding and unembedding matrices) across different models.

They then provide a closed form for the local linear relationship between tokens (basically a vector for indicating how a token embedding can be reconstructed using k-nearest neighbors ) and then compare the cosine similarity of such vectors across models, illustrating a high correlation.

Additionally the authors consider the Intrinsic dimension of a token (extracted based on the k-nearest neighbors) and propose an idea where tokens with lower intrinsic dimension (after preforming PCA on the k-NN matrix) have stronger semantics similarity to their neighbors.

Finally the work continues by exploring the possibility of transfer for steering vectors (used to bias or sway the model's behavior towards some desirable direction) from one model to another (of the same family). The results illustrate how one can use steering vector extracted from smaller model and apply it to larger models (and vice versa).

**Questions To Authors:**

Q1:Does the K value have significant impact for Fig2 ? additionally why use such a large k for the ID (1200) ? seems a bit random and too large

Q2: Any data on what were the hypothetically undertrained tokens  with zero cosine similarity ? did they match the same characteristics of land and Bartolo, 2024

Q3: For the steering and Fig 4, can you please explain how this vector is found ? and why do the authors change the multiplies $\alpha$ for the cases where the model's own steering vectors were used ? Also are there any major differences going from a larger -> smaller model compared to small -> large ? I don't see any in Fig 5 and it is a bit surprising

Q4: For the proof in Appendix A, can the authors explain line 407 ? I am not sure how justify using the $\epsilon \mathbb{I}$ term. I understand why the need for invertibility but I can't justify the definition of $C_i$. Also can't there we use pseudo-inverse or SVD decomposition ?

Q5: In this work the authors claim that the embeddings of the tokens are independent of their contextual usage ? I do find this idea somewhat similar or at times  contrary to works regarding  Neural Collapse in language models. Works such as Wu et. al. 2024 (Linguistic Collapse)  or Zhao et. al 2024 (Implicit Geometry of Next Token Prediction) revolve around such discussions but do operate mostly on the penultimate layer feature embeddings.

Q6: With regards to the steering vector, do the authors have any results applying the transfer on models from different families ? i.e. llama -> gpt ?

**Reasons To Accept:**

Pretty solid paper. I believe the idea and standardization of the study is quite novel and unique. The paper is written well and the illustration are easy to follow. The experiments were quite convincing for me (despite at times making too general of a claim). I will further discuss in questions but I believe this work is a strong accept.

**Reasons To Reject:**

As mentioned some of the claim are a big bold given the results. For example, I am not entirely sure how much I can get behind the ideas presented in section 5.3 and Table 1. While novel, I would like to see some more quantitative analysis perhaps explaining the results and further analyzing the behavior of the Gemma models.

Additionally, I might have misunderstood but at the Discussion phase the authors claim that the hidden state embedding persist throughout the model ? I am not sure but I interpret that as different layer having similar token-wise orientation withing a model ? If so I fail to see where this result was discussed.

Nonetheless I don't believe these are reasons to reject this work, just some points for improvement. I should mention I am not as familiar with the steering vector setup, so I could be missing potential problem there.

---

> ### Author Response · Authors · 2025-06-01
>
> Thank you for your careful review, and encouraging assessment!
>
> Overall, we will make sure our claims are made precise and are not too general.
>
> > ... section 5.3 and Table 1: While novel, I would like to see some more quantitative analysis.
>
> Thank you for the suggestion - please see our response to all reviewers above, where we provide new quantitative evidence to support our claim that embeddings with lower intrinsic dimension have semantically coherent clusters.
>
> > at the Discussion phase the authors claim that the hidden state embedding persist throughout the model ?
>
> We will re-word our claims here - we just meant to say that steering vectors are transferable, suggesting that the model’s hidden states seem to also have similar representations/geometry for the steered concepts.
>
> > Q1:Does the K value have significant impact for Fig2?
>
> We will add additional experiments with additional hyperparameters for k for Figure 2 and Table 1. Overall, with alternative values of k, we observe the same trends in both Figure 2 and Table 1.
>
> In particular, for Table 1, our quantitative experiment using Semantic Coherent Scores (see response to all reviewers above) demonstrates that our findings not only hold across a range of k values, but also lower k values actually lead to better results - so thank you for your question!
>
> We originally started with a much larger k value (1200) that is similar to the dimension of the model’s embedding space (1024). Otherwise, when computing the PCA on the k nearest neighbors, we end up with a [k, d] matrix where k << d, ie, a low rank matrix.
>
> > Q2: Any data on what were the hypothetically undertrained tokens with zero cosine similarity ? did they match the same characteristics of land and Bartolo, 2024
>
> Yes, we observe some overlap in our undertrained tokens and that of Land and Bartolo.
>
> For instance, Land and Bartolo report 2,225 undertrained tokens for Llama3-8B. Our procedure produces 13,967 candidates.
> Our approach matches 924 of their 2,225 tokens, yielding a recall of 0.415 and a precision of 0.066.
> Examples of overlapping tokens include:
> ```
> 'etiyle', '\tTokenNameIdentifier', 'галі', '_REALTYPE', '/contentassist', 'NICALL', 'ดาห', ' fChain', 'รงเร', 'ilmektedir', ' Mühendis', '（笑', 'itempty', ' ediyorum', 'Winvalid', ' ابراه', 'ามารถ', ' коштів', 'GameObjectWithTag', '/settingsdialog', ' профилакти', 'teří', 'นวย', 'HeaderCode', 'อนไลน', 'ایسه', ' přítom', 'ılmış', 'ablytyped', '.bunifuFlatButton'
> ```
>
> Overall, we observe our undertrained tokens yield a lot of non-English tokens.
> We will add a section in the Appendix regarding this observation. Thank you for this suggestion.
>
> > Q3: For the steering and Fig 4, can you please explain how this vector is found ? and why do the authors change the multiplies
>  for the cases where the model's own steering vectors were used ? Also are there any major differences going from a larger -> smaller model compared to small -> large ?
>
> The steering vectors for Llama3 and GPT2 are each derived following prior work. For Llama3, we follow [1], in which they average the difference in mid-layer activations given pairs of positive or negative examples of a target behavior (e.g., sycophancy). For GPT2, we follow [2], in which they train a linear probe model from the model’s hidden states to classify an input as toxic or non-toxic. The resulting probe vector is then used to steer the model.
>
> Multiplying $\alpha$ is a common hyperparameter seen in most steering experiments [1, 2, 3, 4], regardless of whether the steering vector is transferred or not. The idea is that given some steering vector that represents a concept (ie, toxicity), scaling $\alpha$ larger will induce more of the target concept in the model’s hidden representation, and elicit the target behavior more strongly. The same applies to Sparse Autoencoder (SAE) latents - SAE latents are scaled by $\alpha$ to control how strongly they elicit a target behavior.
>
> We did not observe much of a difference in transferring from small to large or large to small language models (as you point out, in Figure 5, but also in Figure 9 of the appendix, where we steer Llama3-3B with steering vectors from 1B and 8B).
>
>
> [1] Panickssery et al. “Steering Llama 2 via Contrastive Activation Addition”. 2024
>
> [2] Lee et al. “A Mechanistic Understanding of Alignment Algorithms: A Case Study on DPO and Toxicity”. 2024
>
> [3] Li et al. “Inference-Time Intervention: Eliciting Truthful Answers from a Language Model”. 2024
>
> [4] Wu et al. “ReFT: Representation Finetuning for Language Models”. 2024

---

> > ### Author Response · Authors · 2025-06-01
> >
> > > Q4: For the proof in Appendix A, can the authors explain line 407 ?
> >
> > Adding $\epsilon I$ is a common trick (ridge regularization, aka Tikhonov regularization), which provides a few benefits but here are two:
> > 1. By defining $C_i = Z_i^TZ_i + \epsilon I$, every eigenvalue of $Z_i^TZ_i$ gets shifted by $\epsilon$, and thus $C_i$ becomes strictly positive definite (becoming invertible), and allowing a unique solution.
> > 2. In line 410, we are interested in minimizing $w^TCw = w^TZ^TZw = ||Zw||^2$. By defining $C = Z^TZ + \epsilon I$, this becomes $||w^TCw||^2 = ||Zw||^2 + \epsilon ||w||^2$, where the first term $||Zw||^2$ is our reconstruction error and the $\epsilon ||w||^2$ term is a ridge penalty term, where the ridge penalty term similarly allows for the optimization objective to have a unique solution.
> >
> > In practice, the use of a pseudo-inverse or SVD is also acceptable and would likely lead to similar results.
> >
> > Lastly, note that LLE (as well as our closed form approach) is an approximate reconstruction of token embeddings to begin with.
> >
> > > Q6: With regards to the steering vector, do the authors have any results applying the transfer on models from different families ? i.e. llama -> gpt ?
> >
> > In this work we did not study cross-family transferability or geometric similarities. This was mostly because our approach requires matching tokens between embedding spaces, and thus requires LMs that share the same tokenizer (ie, same model family).

---

> > ### Comment · Reviewer_PyR3 · 2025-06-08
> >
> > Apologies for the late response. I had to re-evaluate this paper based on some of the responses.
> >
> > While I am happy with the responses to my other questions, I would like to respectfully request that the authors revisit Question 5, which I believe was not adequately addressed in the current draft. After a closer examination of Section 4, I am concerned that the primary findings on “Model Families Share Similar Token Orientations (Usually)” may largely be a byproduct of shared training data across models, rather than an intrinsic or architecture-driven phenomenon. **This shared global geometry regardless of model is discussed in the aforementioned works**.
> >
> > In particular, recent work such as Wu et al. (2024), which explores Neural Collapse in language models, offers a data-driven perspective on embedding geometry. Their framing of next-token prediction as a classification problem — and the finding that last-layer features collapse for contexts sharing the same next token — provides a strong precedent for viewing embedding geometry as emergent from dataset structure. Additionally, Zhao et al. (2024) offers both theoretical and empirical evidence that the geometry of unembedding vectors (referred to in their work as last-layer weights) is tightly linked to the distribution of next-token support sets in the training corpus.
> >
> > While the current paper presents a compelling empirical study, it does not engage with these relevant lines of work. The conceptual overlaps are significant: both Wu et al. and Zhao et al. propose that token representations are shaped by token-level supervision signals and co-occurrence patterns, which is fundamentally the same mechanism that could explain the shared global geometry reported in this paper (different models sizes of the same family most likely had the same training data). In fact, the phenomenon of aligned token embeddings across models is explicitly theorized and empirically analyzed in Zhao et al. (2024) through a optimization framing — yet the authors do not cite or discuss them. I believe a more thorough engagement with this literature is essential to properly contextualize and substantiate the paper’s claims.
> >
> > Given the conceptual proximity between the current work and the recent studies cited above, I find this omission troubling. I understand that the literature in this area is vast and fast-evolving, and it is reasonable that some relevant work may be unintentionally overlooked. However,**I am particularly troubled by the fact that this issue was explicitly raised in my original review as Question 5, yet it was not addressed in the author response at all**. Given how directly related these prior works are to the claims made in this paper, I believe a thoughtful engagement with them is necessary. As it stands, given that this global geometry of LLMs is discussed in other forms in previous work, it takes away from some of the novelty of the paper regarding this particular topic. One of the reasons that I assigned a high score to this work was because I was unsure of the similarity between the paper and previous literature and still found the idea to be novel. Given the lack of discussion, I am reducing my score to a 6.

---

> ### Author Response · Authors · 2025-06-08
>
> Apologies for omitting a response to question 5 - We genuinely did not understand the question. We will read the two papers tonight and respond soon.
>
> In the meantime, can you clarify what you mean by "[authors] claim that token embeddings are independent of their contextual usage"? I'm not sure we ever made such a claim, and I'm not sure what exactly is being asked in question 5?
>
> Thank you!

---

> > ### Comment · Reviewer_PyR3 · 2025-06-08
> >
> > I am happy to elabotrate. If I have the same token, in different sequences, the embedding vector for that token is still the same I assume.  this is what I was referring to as the context independance of token embeddings.
> >
> > This is not true for the unembedding given that for a token token $t_j$, the unembedding vector $h_{j}^{L-1}$ depends on the tokens appearing before it in the sequence $[t_0, t_2,  \dots , t_{j-3}, t_{j-2}, t_{j-1} ]$.

---

> > ### Author Response · Authors · 2025-06-09
> >
> > Thank you for your response, and for the two pointers. Both papers were great!
> >
> > > This is not true for the unembedding given that for a token $t_j$, the unembedding vector $h^{L-1}$...
> >
> > We first wish to quickly clarify a minor point regarding your response above regarding “unembedding vector $h^{L-1}$” – although it’s possible that we’re just confusing ourselves with notations, but hopefully this also clarifies the notations used in the rest of our response below.
> >
> > After going through $L$ transformer blocks, LMs build a context representation in its penultimate layer ($\mathbf{h}^{L-1} \in \mathcal{R}^d$). This context representation is mapped back to a “vocabulary space” using unembedding vectors, $\mathbf{U} \in \mathcal{R}^{V \times d}$. The softmax on the result ($\mathbf{Uh}^{L-1} \in \mathcal{R}^V$) predicts the next token. So, while the argument to the softmax ($\mathbf{Uh}^{L-1}$) is context dependent, the unembedding weights $\mathbf{U}$ themselves are not.
> >
> > Now indeed, there is an interesting, symmetric relationship between context representations ($\mathbf{h}^{L-1}$) and the unembedding layer ($\mathbf{U}$). As Zhao et al. puts it, $\mathbf{h}^{L-1}$ and $\mathbf{U}$ can be thought of as a log-bilinear model. We provide a similar characterization in Equation 6. Perhaps this symmetric, bilinear relationship is what you mean by unembedding vectors being context dependent.
> >
> > Indeed, the two works you point to study the geometry of either the penultimate layer $\mathbf{h}^{L-1}$ or the bilinear model $\mathbf{Uh}^{L-1}$. First, regarding neural collapse, it is neat to see neural collapse studied in language models! There can definitely be an interesting relationship between neural collapse and geometry of unembeddings. Namely, if we put aside for a moment the strict criteria for neural collapse (ex: simplex ETF) and simply think of neural collapse as an “optimal representation” that can be learned in $\mathbf{h}^{L-1}$, then it is certainly expected that models that reach neural collapse will also have similar geometry in their symmetric counterpart, $\mathbf{U}$. It would be interesting to empirically confirm this – that models with stronger signatures of neural collapse will also have more similarities in the geometric signatures that we study.
> >
> > Regarding Zhao et al., there are definitely strong connections between their theoretical findings and our work. For instance, their finding that $\mathbf{Uh}^{L-1}$ will be dominated by a low-rank component, driven by word co-occurrences, could be a potential explanation for the similarity of  geometries seen in our work. Finding further connections between the two works could be a great contribution.
> >
> > Yes, there is a close relationship between the geometry of context representations ($\mathbf{h}^{L-1}$) and unembeddings ($\mathbf{U}$). Yes, all 3 papers study the similarities in the geometry of either $\mathbf{h}^{L-1}$, $\mathbf{Uh}^{L-1}$, or $\mathbf{U}$.
> >
> > However, some specific contributions in our work include:
> > * Simple ways of characterizing and comparing global and local geometry (ex: LLE)
> > * Relationship between low intrinsic dimensions and semantically coherent clusters
> > * Transferability of steering vectors
> >
> >
> > Lastly, regarding omitting a response to question 5 - genuinely, there was no malicious intent. We did not understand what was meant by “[authors] claim that token embeddings are independent of contextual usage” and how it connects to the two papers. Again, in discussing why steering vectors are transferable (Section 6, Equation 6), we discuss the bilinear relationship between $\mathbf{h}^{L-1}$ and $\mathbf{U}$, just like Zhao et al.’s log-bilinear setup, and even use this argument in our response to reviewer tRQC for why untied models have dissimilar embeddings but similar unembeddings.
> >
> > (Realizing as we write this up now - we can see how our response to reviewer tRQC regarding the bilinear relationship in $\mathbf{h}$ and $\mathbf{U}$ combined with the omission to question 5 could have looked suspicious - maybe this is what you are referring to.)
> >
> > For what it’s worth, we are used to rebuttals having character limits (ex: 6,000 characters for Neurips), which historically meant that one could not always respond to every point and had to prioritize what to respond to. While COLM does not have a character limit, the instructions specifically say “Regarding length: we **strongly recommend against** writing long responses. Please be succinct and to-the-point in your response. This lowers the barrier for reviewers to engage meaningfully.” While COLM is new and thus does not have established norms yet, we took this to mean that we should follow the norms of other ML conferences.
> >
> > All of this is to say, we genuinely did not mean harm and apologize for the misunderstanding. We kindly ask that your score reflects what you view as our contributions (in light of **all** relevant prior work), and not any misunderstandings.
> >
> > Thank you!

---

> > > ### Comment · Reviewer_PyR3 · 2025-06-09
> > >
> > > Thank you for the response. I would like to mention that  I agree with the authors on how the difference in notation contributed to some misunderstanding. I was reviewing during a more busy period and I may have not articulated myself as well as I could have. However, I would have been happy to respond to any followup question to clear confusion. I find this work genuinely interesting, which is reflected in my score and why I care to have a deeper discussion on the topic. Unless I am missing something about your results, the depth of the connection to Zhao et al. might indeed be deeper:
> > >
> > > My understanding is they characterize the geometry (in terms of gram matrices) of both context and word (un)embeddings. While this connects to the neural collapse geometry, it extends beyond a simple simplex ETF geometry: this geometry is implicitly determined by the SVD factorization of a data-sparsity matrix derived from the training data. Their key point that I find particularly relevant to your findings is that this implies that for sufficiently expressive models, the very geometry of both context and word embeddings is dictated by the training data itself, largely independent of specific architectural nuances.
> > >
> > > Therefore, again please correct me if I miss something, but when the authors observe ‘high correlations in the geometry of word embeddings’ across different models of the same family, this is not merely an interesting relationship but rather a direct empirical manifestation of the theoretical framework laid out by Zhao et al. Their work specifically shows that WTW (or UTU in the authors’ notation for word unembeddings) becomes a function of the training data in such models. Granted, their experimental analysis is done in a more controlled environment compared to the general LLM results provided in your work.
> > >
> > > It also seems like some of the findings on steering vectors and semantic clustering largely reflect properties of the embedding geometry, which makes the connection to prior work on LLMs feature geometry even more interesting.

---

> > > > ### Author Response · Authors · 2025-06-09
> > > >
> > > > We certainly share your hypothesis that data is likely the largest culprit in the geometric similarities, as this is essentially the key argument for the “Platonic Representation Hypothesis”. We have some preliminary (weak) evidence for this argument.
> > > >
> > > > In particular, we finally found two models from **different** families that share the same tokenizer. GPT-NeoX-20B has been trained on a dataset called the Pile, while Olmo-7B has been trained on a dataset called Dolma. Luckily, they share the same tokenizer, allowing us to run our analysis.
> > > >
> > > > We ran our global geometry similarity analysis from 4.1 (Figure 1) on their *unembeddings*. The two models have a Pearson correlation score of 0.32 -- not as strong as within-family models, but still has some weak correlation, likely because of similar language usage in the two datasets for some tokens. We only state this as *preliminary, weak* evidence because we don’t yet know how much of this lower alignment can be attributed to the difference in data versus the difference in model architectures, but we speculate that the difference in data plays a bigger role.
> > > >
> > > > We think there are many exciting future directions, given these open source datasets (Pile, Dolma). For instance, a deeper dive into the statistics (eg, co-occurrence) in the data and how they relate to the resulting geometry could be a way to close the gap between theory and practice.
> > > >
> > > > Although this finding could be viewed as “an instantiation of existing theory”, we hope you still view these findings as good science and meaningful contributions.

---

> > > > > ### Comment · Reviewer_PyR3 · 2025-06-10
> > > > >
> > > > > Thank you. Just to clarify, this is not my hypothesis, but rather the claim of Zhao et al. that I am bringing up. Regarding the experiment that you mention, from what I know, the Pile is still only a small subset of Dolma, so the fact that the correlation is 0.36 does not refute or validate the claim of Zhao et al. Instead, it remains pretty clear to me, and you don’t seem to object, that your paper’s finding that models trained on the same data share the same geometry of unembedding representations across various sizes, is aligned with their claim. Thus, their analysis provides a direct possible justification of your findings . Without undermining your finding, I believe stating this explicit connection in your paper is useful and imprortant.
> > > > >
> > > > > In light of all this discussion, I have raised my score. Good luck.

---

### Official Review · Reviewer_tRQC · 2025-05-13

**Rating:** 7
**Confidence:** 4
**Ethics Flag:** 1

**Summary:**

This paper explores the shared geometric structure of token embeddings across language models. It identifies global similarities in relative orientations and local similarities through two methods: Locally Linear Embeddings (LLE) and a new intrinsic dimension (ID) metric. Tokens with lower IDs often form semantically coherent clusters, suggesting low-dimensional manifolds. The study further shows that alignment extends into hidden states, enabling steering vectors to be transferred across models with different architectures.

=== After Rebuttal ===

Given authors' responses, I have decided to revise my rating from 4 to 7.

**Questions To Authors:**

1. What is the interpretation of the discrepancy between global and local similarity in models like LLaMA-1B vs. LLaMA-8B?

2. Have you evaluated the geometric similarity across families (e.g., LLaMA vs. GPT2), especially since the steering vector transfer spans these models?

**Reasons To Accept:**

1. The work explores an interesting topic in the interpretability and geometric analysis of LLM embeddings.

2. Clear presentation of preliminaries, formulas, and notations.

**Reasons To Reject:**

1. The paper lacks high-level visualizations or concrete examples that could help readers intuitively understand the observed geometric structures.

2. The paper would benefit from more high-level interpretations and discussions. For example, in Lines 131–133, what are the possible causes or explanations that untied language models exhibit high unembedding correlations but low embedding correlations?

3. There is an inconsistency in the results: models like LLaMA-1B and 8B show high local structure similarity but low global structure similarity (e.g., Figure 1 vs. Figure 2). These findings are not well-addressed or explained. (Also see question 1.)

4. Misalignment Between Findings and Applications: Most geometric similarity analyses are conducted within the same model family (e.g., GPT2 models or LLaMA variants), but the application section jumps across families (e.g., steering between LLaMA and GPT2) without showing the corresponding geometric similarity between these cross-family pairs.

---

> ### Author Response · Authors · 2025-06-01
>
> Thank you for your careful review!
>
> > The paper lacks high-level visualizations
>
> We can add a high-level visualization to our revision, but do not view this as a reason for rejection.
>
> > The paper would benefit from more high-level interpretations and discussions. For example, in Lines 131–133, what are the possible causes or explanations that untied language models exhibit high unembedding correlations but low embedding correlations?
>
> Thank you for the suggestion. We will add a high-level discussion on why we think untied language models demonstrate similar global structure in the unembedding space, but not the embedding space.
>
> Here is our intuition: following notations from Section 3, the language model first assigns each token an embedding, using matrix $E$, which becomes the first hidden state $\mathbf{h}^0$.
> The final step projects the last hidden state onto the unembedding space: $〈\mathbf{h}^{L-1}, U〉$.
>
> Figure 1 tells us that for Llama, their unembeddings $U$ (and thus last layer hidden states $\mathbf{h}^{L-1}$) converge across models, while embeddings do not.
>
> An intuitive way to understand this is that across Llama models, token embeddings start off in a different place in its $R^d$ space, but after going through L transformer blocks, their final locations in unembedding space converge across LMs.
> Put differently, regardless of where the embeddings start (ie., $E$), transformer hidden states construct similar representations ($\mathbf{h}^{L-1}$), and converge to learning a similar unembedding space $U$.
> Where the embeddings start (ie., $E$) does not seem to matter, as they end up looking similar by the time they reach the last layer.
> It is essentially the final representations as well as the unembeddings that become similar – in some LMs, the fact that embeddings also share similar geometry is only because the embeddings and unembeddings are tied.
>
> We think the implications of tied versus untied embeddings are underexplored, and view this space as an interesting direction for future work.
>
> > There is an inconsistency in the results (Figure 1 vs. 2)
>
> We believe there may be a misunderstanding. **Figure 1 and 2 do not have a discrepancy.**
>
> Both suggest that Llama models demonstrate both global and local similarity in the **unembedding** space. Your point regarding the low global similarity of Llama-1B vs. 8B have a Pearson correlation score of 0.87 (Figure 1), suggesting high global similarity.
>
> One possible source of confusion is that in Figure 2, we are looking at the **unembeddings** of Llama, not the embeddings. We will clarify the caption of Figure 2 to make this more explicit (The main text, line 152, notes that Figure 2 is the unembeddings of Llama).
> Furthermore, Figures 6, 7, 8 in the Appendix show results of local geometry for other LMs. In particular, Figure 7 shows results for the **embeddings** of Llama3, which demonstrates lower local similarity, which demonstrates consistent results as Figure 1.
>
> **In summary, there is no discrepancy between Figure 1 and Figure 2.**
>
> > Misalignment Between Findings and Applications
>
> We believe there is a misunderstanding. In our application section, we **do not** transfer steering vectors across model families. In Figure 4, we steer Llama3-8B with steering vectors from Llama3-1B and 3B (stays within Llama family). In Figure 5, we steer GPT2-{S,M,L,XL} with steering vectors from another GPT2 model (stays within GPT2-family).
>
> **In summary, there is no misalignment between our findings and our applications.**
>
> > What is the interpretation of the discrepancy between global and local similarity in models like LLaMA-1B vs. LLaMA-8B?
>
> Please see our response above - there is no discrepancy in global vs local similarity in Llama-1B vs. 8B.
>
> > Have you evaluated the geometric similarity across families (e.g., LLaMA vs. GPT2)
>
> We have not studied geometric similarities across families. This is mainly because we require matching token embeddings across embedding spaces, and for matching tokens to exist, the LMs must use the same tokenizers.
>
>
> If there are any other concerns or questions, please let us know. If some confusions have been clarified, we kindly ask the reviewer to consider updating their score.

---

> > ### Comment · Reviewer_tRQC · 2025-06-03
> > **Re: Official Comment by Authors**
> >
> > Thank you for the detailed responses and clarification, and apologies for my earlier misunderstanding. Most of my previous concerns have now been addressed. I also appreciate the authors’ willingness to incorporate more high-level explanations in the revision. Given these clarifications and the authors' intent to improve the presentation and discussion in the revision, I have decided to revise my rating from 4 to 7.

---

### Official Review · Reviewer_xUSm · 2025-05-13

**Rating:** 7
**Confidence:** 4
**Ethics Flag:** 1

**Summary:**

This paper investigates the geometric structure of token embeddings across different LLMs. The authors identify two key properties: global similarity, where token embeddings share relative orientations across models, and local similarity, based on both Locally Linear Embeddings (LLE) and intrinsic dimension (ID). The study shows that many tokens lie on low-dimensional manifolds, and those with lower ID often form semantically coherent clusters. The paper shows that steering vectors, i.e. interventions used to guide model behavior, can be transferred across models due to shared embedding geometry, enabling novel interpretability applications.

**Reasons To Accept:**

- evidence that LLMs share both global and local geometric embedding structures which is underexplored in current literature.
- extensive experiments across multiple model
- use of Pearson correlation, LLE, and intrinsic dimension are intuitive and well-motivated
- finding that steering vectors can be transferred between models is both surprising and practically valuable for model alignment and control.

**Reasons To Reject:**

-  the paper lacks deeper theoretical analysis of why the alignment and transferability emerge.
- The semantic coherence of low-ID clusters is shown only via qualitative examples

---

> ### Author Response · Authors · 2025-06-01
>
> Thank you for your careful review and your encouraging assessment!
>
> > the paper lacks deeper theoretical analysis of why the alignment and transferability emerge.
>
> We certainly agree that adding a theoretical account would strengthen our work, but view it as outside the scope of this empirical paper. Alternative studies could include studying the learning dynamics of language models during pre-training, and studying how the geometry of token embeddings evolve.
>
> > The semantic coherence of low-ID clusters is shown only via qualitative examples
>
> Please see our response to all reviewers above where we provide quantitative evidence that embeddings with lower intrinsic dimension have semantically coherent clusters.

---

> > ### Comment · Reviewer_xUSm · 2025-06-04
> >
> > Thank you for your response. Everything looks good to me. I believe this is a strong paper and I will maintain my current score.

---

### Author Response · Authors · 2025-06-01

Dear all reviewers,

Thank you for your careful review. We believe your input has improved our work.

Here we provide a response to a common point mentioned by multiple reviewers.
Namely, regarding our finding that embeddings with lower intrinsic dimension (ID) have semantically coherent clusters, both reviewers xUSm and PyR3 noted that we only demonstrate this with qualitative examples.

To this point, we provide quantitative evidence below, which we will add to our next revision.

Namely, we design a “semantic coherence score” (SCS) and measure the SCS of 500 randomly sampled token embeddings.
To measure SCS, for each token $x$, we look at its k (=50) nearest neighbors in embedding space ({$x_k$}) and compute the shortest distances from token $x$ to tokens $x_k$ in ConceptNet.
ConceptNet is a knowledge graph of 34 million nodes and edges, each node representing a concept (token) and edges representing one of 34 relation types.

Formally, for each token $x$ and its k nearest neighbors, we measure the average shortest path from $x$ to $x_k$ in ConceptNet, normalized by some cutoff threshold L (=5):

$\text{SCS}(x) = 1 - \frac{1}{k}\sum_{i=0}^{k-1}\hat{d}(x, x_i)$

$\hat{d}(x, y) = \frac{min(d(x, y), L)}{L}$

where d(x, y) = shortest path from token x to token y in ConceptNet

SCS takes a value between 0~1. A score of 0 means that x is not connected to any of its k neighbors within L hops, while a score of 1 means x is connected to all k neighbors within L hops.

Finally, we measure the Spearman correlation between the SCS of tokens and their intrinsic dimensions.
Our correlation scores and p-values are as follows  (k here indicates the hyperparameter for computing intrinsic dimension (number of nearest neighbors used to calculate ID), not to be confused with the k used to calculate SCS):

| Model | Spearman Correlation | P-Value |
| :---------                       | ------: | --------: |
| GPT2 (k=100)               | -0.67 |     1e-66 |
| GPT2 (k=300)               | -0.63 |     2e-56 |
| GPT2 (k=500)               | -0.59 |     6e-48 |
| GPT2 (k=1000)             | -0.56 |     5e-42 |
| Llama3-3B (k=100)      | -0.45 |     7e-27 |
| Llama3-3B (k=300)      | -0.46 |     1e-27 |
| Llama3-3B (k=500)      | -0.37 |     8e-18 |
| Llama3-3B (k=1000)    | -0.18 |       4e-5 |
| Gemma2-3B (k=100)   |  0.22 |       2e-7 |
| Gemma2-3B (k=300)   |    0.1 |       0.02 |
| Gemma2-3B (k=500)   |  0.03 |         0.5 |
| Gemma2-3B (k=1000) |  0.02 |         0.6 |

Across a range of k values, GPT2 and Llama3 demonstrate high correlations.
We will also add figures (scatterplots) plotting the intrinsic dimension vs. semantic coherence scores.
Here is an example for GPT2: https://imgur.com/a/eujmPrp

We observe that we do not see any correlation in Gemma2 - we conjecture that this is because of the high number of non-English tokens, special tokens, or emoji-like tokens in Gemma2, which ConceptNet does not handle very well (Gemma2 has a vocab size of 256k, as opposed to 128k in Llama3 and 50k in GPT2).

---

### Decision · Program_Chairs · 2025-07-08

**Decision:**

Accept

**Comment:**

This is a strong, well-supported paper that provides valuable and actionable insight into what is being learned by transformers. There is positive engagement by the reviewers on a few issues that could improve presentation quality, but all are supportive.